# Synthesis of Novel *α*-Trifluorothioanisole Derivatives Containing Phenylpyridine Moieties with Herbicidal Activity

**DOI:** 10.3390/molecules27185879

**Published:** 2022-09-10

**Authors:** Zengfei Cai, Wenliang Zhang, Zhongjie Yan, Xiaohua Du

**Affiliations:** 1Catalytic Hydrogenation Research Center, Zhejiang Key Laboratory of Green Pesticides and Cleaner Production Technology, Zhejiang Green Pesticide Collaborative Innovation Center, Zhejiang University of Technology, Hangzhou 310014, China; 2Agrowin (Ningbo) Bioscience Co., Ltd., Ningbo 315100, China

**Keywords:** synthesis, *α*-trifluorothioanisole derivatives, phenylpyridine, herbicidal activity

## Abstract

To discover novel herbicidal compounds with favorable activity, a range of phenylpyridine-moiety-containing α-trifluorothioanisole derivatives were designed, synthesized, and identified via NMR and HRMS. Preliminary screening of greenhouse-based herbicidal activity revealed that compound **5a** exhibited >85% inhibitory activity against broadleaf weeds *Amaranthus retroflexus*, *Abutilon theophrasti*, and *Eclipta prostrate* at 37.5 g a.i./hm^2^, which was slightly superior to that of fomesafen. The current study suggests that compound **5a** could be further optimized as an herbicide candidate to control various broadleaf weeds.

## 1. Introduction

Organosulfur compounds are widely present in animals and plants due to their physiological activities, such as diallyl disulfide compounds in garlic, isothiocyanate compounds in cruciferous vegetables and some fruits, and sulfur-containing amino acids and vitamins. These molecules play a vital role in maintaining the metabolism of organisms and various life activities. Therefore, many researchers have conducted in-depth research on these compounds and found that some molecules containing thioheterocycles, thioureas, sulfonamides, thioethers, sulfoxides, sulfones, and other structures show biological activity [1,2,3,4,5,6,7,8]. For instance, Wang et al. [9] reported a class of sulfonylurea compounds with high herbicidal activities against *Echinochloa crusgalli* and *Digitaria sanguinalis* for pre- and post-emergence treatment and noted that they were safe for peanut as post-emergence treatment. A range of 5-substituted sulfonylurea derivatives discovered by Li et al. [10] showed good herbicidal activities against *Amaranthus retroflexus* and *Brassica campestris* for pre-emergence treatment. It is worth noting that compounds containing sulfide, sulfoxide, and sulfone structures have been widely used in materials [11,12], medicine [13,14,15], and pesticides [16,17,18,19,20,21,22] in the past few decades (Figure 1). The pre-emergence herbicide pyroxasulfone, discovered by Todoroki et al. [23], exhibited excellent herbicidal activity and crop safety, and it is currently used in the field.

Substituted 2-phenylpyridines discovered by Schaefer et al. exhibited good inhibition activity against weeds [24,25]. Substituted 3-(pyridin-2-yl)benzenesulfonamide derivatives disclosed by Liu et al. showed excellent inhibitory activity against a variety of weeds [26,27,28]. Du et al. also reported that a range of kresoxim-methyl derivatives containing phenylpyridine moieties exhibited higher inhibitory activities against broadleaf weeds than mesotrione [29,30].

Herein, 18 novel *α*-trifluorothioanisole derivatives were synthesized through introducing phenylpyridine moieties into *α*-trifluorothioanisole. The inhibitory activities of the resultant compounds against broadleaf and grass weeds were determined.

## 2. Results and Discussions

### 2.1. Chemistry

The synthesis procedures for the target compounds used in this work are outlined in Figure 1. Intermediates **3a–3f** were obtained according to the literature [31]. The target compounds **5a–5f** were prepared via nucleophilic substitution reaction from intermediates **3a–3f** and compound **4**. In addition, compounds **5a–5f** were oxidized to yield compounds **6a–6f** and **7a–7f** using 3-chloroperbenzoic acid or hydrogen peroxide as oxidants according to previously disclosed methods [32,33]. After synthesis, all target compounds were characterized via HRMS and NMR. X-ray diffraction crystallography was further used to confirm the structure of compound **5a** (Figure 2). The crystal data of compound **5a** and the NMR spectrum of all target compounds are shown in the Appendix A.

### 2.2. Greenhouse Herbicidal Activity Assays

As can be seen from Table 1, some target compounds exhibited excellent inhibitory activity against the tested dicotyledonous weeds but proved ineffective against monocotyledonous weeds. Of these, at 150 g a.i./hm^2^, compounds **5a**, **5f**, **6b**, and **7a** exhibited >80% inhibitory activity when used for the post-emergence treatment of the broadleaf weeds *AT*, *AR*, and *EP*, while compounds **6f** and **7f** exhibited >80% activity against *AT* and *AR*. Furthermore, compound **5a** also effectively suppressed the growth of *DS* and *SV*, which was slightly better than the positive control fomesafen. Other compounds exhibited varying levels of general herbicidal activity. Further analysis revealed that compound **5a** exhibited >85% inhibition against *AT*, *AR*, and *EP* for post-emergence treatment at 37.5 g a.i./hm^2^, which was slightly superior to the inhibition ability of fomesafen, whereas compound **5f** also exhibited >70% inhibition against these three weeds.

From Table 1, we can see that the herbicidal activities of compound **5** were slightly better than those of compounds **6** and **7**. According to the SAR of compound **5** in the field of herbicidal activity, when R_1_ and R_2_ were both substituted by hydrogen atoms (**5a**), the activities of compound **5** against three broadleaf weeds were better than those of other compounds. For compound **6**, when R_1_ was substituted by a hydrogen atom and R_2_ was substituted by a fluorine atom (**6b**), the herbicidal activities of compound **6** were optimal. For compound **7**, the optimal herbicidal activities were observed when R_1_ and R_2_ were substituted by either hydrogen atoms or fluorine atoms (**7a**, **7f**).

## 3. Materials and Methods

### 3.1. Instrumentation

All reagents and other materials were purchased from commercial sources and used without additional purification unless otherwise noted. A B-545 melting point instrument was used to determine melting point without calibration. A Bruker AV-400 or AV-500 MHz spectrometer was used to generate NMR spectra with DMSO-d_6_ or CDCl_3_ serving as solvents. An Agilent 6545 Q-TOF LCMS spectrometer was used for mass spectrometry. A Bruker D8 Venture diffractometer was utilized to collect crystallographic data.

### 3.2. Synthesis

The synthesis methods of the title compounds are outlined in Figure 1.

#### 3.2.1. General Approach to the Synthesis of Compounds **3a–3f**

2,3-Dichloro-5-(trifluoromethyl)pyridine **1** (5 mmol), triphenylphosphorus (0.5 mmol), K_2_CO_3_ (10 mmol), substituted p-hydroxybenzeneboronic acid **2a–2f** (5.5 mmol), and palladium(II) acetate (0.25 mmol) were mixed and stirred for 6 h with CH_3_CN (10 mL) and CH_3_OH (5 mL) at 50 °C under N_2_. Thereafter, the mixture was extracted using ethyl acetate (30 mL × 3), rinsed using brine, and concentrated. The remaining residue was then recrystallized using ethanol and water as solvents at 70 °C to obtain compounds **3a–3f** [31].

#### 3.2.2. General Approach to the Synthesis of Compounds **5a–5f**

Substituted phenylpyridines **3a–3f** (2 mmol), *N*,*N*-dimethylformamide (10 mL), and NaH (3 mmol, 0.12 g) were mixed and stirred at 20 °C for 30 min under N_2_. Next, 4-trifluoromethylthiobenzyl bromide (2.4 mmol) was added and stirred for 8 h at 60 °C. The mixture was extracted thrice using ethyl acetate (30 mL × 3), rinsed using brine, and concentrated. Residues were then purified via silica gel column chromatography using ethyl acetate (EA) and petroleum ether (PE) (V_EA_:V_PE_=1:15) to obtain compounds **5a–5f [31]**.

#### 3.2.3. General Approach to the Synthesis of Compounds **6a–6f**

Compounds **5a–5f** (0.5 mmol), 85% 3-chloroperbenzoic acid (0.5 mmol), and dichloromethane (10 mL) were mixed and stirred at 20 °C for 6 h. The mixture was evaporated to remove the solvent. Residues were then purified via silica gel column chromatography using ethyl acetate (EA) and petroleum ether (PE) (V_EA_:V_PE_=1:7) to obtain compounds **6a–6f [32]**.

#### 3.2.4. General Approach to the Synthesis of Compounds **7a–7f**

Compounds **5a–5f** (0.4 mmol), 30% hydrogen peroxide (1.6 mmol), and trifluoroacetic acid (2 mL) were mixed and stirred at 60 °C for 6 h. Thereafter, the mixture was made alkaline using a sodium hydroxide solution, extracted thrice using ethyl acetate (30 mL × 3), rinsed using Na_2_SO_3_, and concentrated. Residues were then purified via silica gel column chromatography using ethyl acetate (EA) and petroleum ether (PE) (V_EA_:V_PE_=1:10) to obtain compounds **7a–7f [33]**.

*3-chloro-5-(trifluoromethyl)-2-(4-((4-((trifluoromethyl)thio)benzyl)oxy)phenyl)pyridine (**5a**)*: White solid, yield 78.8%, m.p. 93.2–95.4 °C. ^1^H NMR (500 MHz, DMSO-d_6_) δ: 8.99 (s, 1H), 8.51 (s, 1H), 7.76 (dd, J = 8.4, 3.4 Hz, 4H), 7.64 (d, J = 8.1 Hz, 2H), 7.17 (d, J = 8.8 Hz, 2H), 5.29 (s, 2H). ^13^C NMR (126 MHz, DMSO-d_6_) δ: 159.17, 158.79, 144.27 (q, J = 4.0 Hz), 140.63, 136.27, 135.71 (q, J = 3.1 Hz), 131.02, 129.57 (q, J = 308.4 Hz), 129.37, 129.17, 128.82, 124.39 (q, J = 33.0 Hz), 122.86 (q, J = 273.4 Hz), 122.29 (q, J = 2.0 Hz), 114.37, 68.43. HRMS (ESI): calculated for C_20_H_13_ClF_6_NOS [M+H]^+^ 464.0305, 466.0276, found 464.0308, 466.0277.

*3-chloro-2-(3-fluoro-4-((4-((trifluoromethyl)thio)benzyl)oxy)phenyl)-5-(trifluoromethyl)pyridine (**5b**)*: White solid, yield 85.9%, m.p. 63.5–65.0 °C. ^1^H NMR (500 MHz, DMSO-d_6_) δ: 9.01 (d, J = 0.9 Hz, 1H), 8.55 (s, 1H), 7.79 (d, J = 8.1 Hz, 2H), 7.70–7.64 (m, 3H), 7.61 (d, J = 8.6 Hz, 1H), 7.41 (t, J = 8.7 Hz, 1H), 5.38 (s, 2H). ^13^C NMR (126 MHz, DMSO-d_6_) δ: 157.57, 150.96 (d, J = 244.7 Hz), 147.15 (d, J = 10.6 Hz), 144.33 (q, J = 3.9 Hz), 140.06, 136.34, 135.89 (q, J = 3.3 Hz), 129.79 (d, J = 6.7 Hz), 129.58 (q, J = 308.3 Hz), 129.42, 128.92, 126.27 (d, J = 3.2 Hz), 124.85 (q, J = 33.1 Hz), 122.80 (q, J = 273.4 Hz), 122.59 (q, J = 1.9 Hz), 117.22 (d, J = 20.0 Hz), 114.71, 69.36. HRMS (ESI): calculated for C_20_H_10_ClF_7_NOS [M−H]^−^ 480.0065, 482.0036, found 480.0064, 482.0043.

*3-chloro-2-(3-chloro-4-((4-((trifluoromethyl)thio)benzyl)oxy)phenyl)-5-(trifluoromethyl)pyridine (**5c**)*: White solid, yield 94.0%, m.p. 82.0–83.5 °C. ^1^H NMR (400 MHz, DMSO-d_6_) δ: 9.02 (s, 1H), 8.57 (s, 1H), 7.88 (s, 1H), 7.78 (t, J = 10.5 Hz, 3H), 7.67 (d, J = 7.2 Hz, 2H), 7.40 (d, J = 8.3 Hz, 1H), 5.41 (s, 2H). ^13^C NMR (101 MHz, DMSO-d_6_) δ: 157.49, 154.37, 144.38 (q, J = 3.9 Hz), 140.14, 136.38, 135.89 (q, J = 3.1 Hz), 130.94, 130.25, 129.68, 129.60 (q, J = 309.1 Hz), 129.45, 128.62, 124.87 (q, J = 33.4 Hz), 122.81 (q, J = 274.1 Hz), 122.48, 121.21, 113.62, 69.28. HRMS (ESI): calculated for C_20_H_10_Cl_2_F_6_NOS [M−H]^−^ 495.9770, 497.9740, found 495.9767, 497.9740.

*3-chloro-2-(3-nitro-4-((4-((trifluoromethyl)thio)benzyl)oxy)phenyl)-5 (trifluoromethyl)pyridine (**5d**)*: White solid, yield 86.5%, m.p. 114.5–116.5 °C. ^1^H NMR (400 MHz, DMSO-d_6_) δ: 9.06 (s, 1H), 8.63 (s, 1H), 8.35 (d, J = 2.2 Hz, 1H), 8.11 (dd, J = 8.8, 2.2 Hz, 1H), 7.80 (d, J = 8.1 Hz, 2H), 7.65 (d, J = 8.1 Hz, 2H), 7.61 (d, J = 8.9 Hz, 1H), 5.51 (s, 2H). ^13^C NMR (101 MHz, DMSO-d_6_) δ: 156.76, 151.71, 144.63 (q, J = 3.6 Hz), 139.64, 138.99, 136.51, 136.12 (q, J = 3.5 Hz), 135.49, 129.74, 129.66 (q, J = 308.8 Hz), 129.27, 128.63, 126.42, 125.33 (q, J = 33.3 Hz), 122.83 (q, J = 274.1 Hz), 122.66 (q, J = 1.9 Hz), 115.28, 69.85. HRMS (ESI): calculated for C_20_H_10_ClF_6_N_2_O_3_S [M−H]^−^ 507.0010, 508.9981, found 507.0010, 508.9983.

*3-chloro-5-(trifluoromethyl)-2-(3-(trifluoromethyl)-4-((4 ((trifluoromethyl)thio)benzyl)oxy)phenyl)pyridine (**5e**)*: White solid, yield 93.4%, m.p. 111.6–113.5 °C. ^1^H NMR (400 MHz, DMSO-d_6_) δ: 9.07 (s, 1H), 8.62 (s, 1H), 8.13 (d, J = 8.8 Hz, 1H), 8.10 (s, 1H), 7.83 (d, J = 8.0 Hz, 2H), 7.67 (d, J = 8.1 Hz, 2H), 7.55 (d, J = 8.7 Hz, 1H), 5.52 (s, 2H). ^13^C NMR (101 MHz, DMSO-d_6_) δ: 157.42, 156.64, 144.47 (q, J = 3.6 Hz), 139.94, 136.39, 135.93 (q, J = 3.5 Hz), 135.35, 129.60 (q, J = 308.7 Hz), 129.51, 129.06, 128.33, 128.15 (q, J = 5.1 Hz), 125.02 (q, J = 33.3 Hz), 123.45 (q, J = 273.6 Hz), 122.80 (q, J = 274.1 Hz), 122.48 (q, J = 1.8 Hz), 117.04 (q, J = 30.7 Hz), 113.76, 69.21. HRMS (ESI): calculated for C_21_H_10_ClF_9_NOS [M−H]^−^ 530.0033, 532.0004, found 530.0031, 532.0010.

*3-chloro-2-(2,3-difluoro-4-((4-((trifluoromethyl)thio)benzyl)oxy)phenyl)-5-(trifluoromethyl)pyridine (**5f**)*: White solid, yield 80.2%, m.p. 79.5–82.0 °C. ^1^H NMR (500 MHz, CDCl_3_) δ: 8.84 (d, J = 1.1 Hz, 1H), 8.06 (d, J = 1.5 Hz, 1H), 7.71 (d, J = 8.2 Hz, 2H), 7.53 (d, J = 8.3 Hz, 2H), 7.19–7.15 (m, 1H), 6.92–6.89 (m, 1H), 5.26 (s, 2H). ^13^C NMR (126 MHz, CDCl_3_) δ: 155.42, 149.29 (dd, J = 252.8, 11.0 Hz), 149.28 (dd, J = 8.2, 2.8 Hz), 144.45 (q, J = 3.6 Hz), 141.76 (dd, J = 249.8, 14.4 Hz), 139.03, 136.78, 134.93 (q, J = 3.2 Hz), 132.36, 129.68 (q, J = 308.6 Hz), 128.26, 127.20 (q, J = 33.9 Hz), 124.84 (t, J = 3.6 Hz), 124.65, 122.74 (q, J = 273.6 Hz), 120.22 (d, J = 12.2 Hz), 110.36 (q, J = 2.6 Hz), 70.97. HRMS (ESI): calculated for C_20_H_9_ClF_8_NOS [M−H]^−^ 497.9971, 499.9942, found 497.9969, 499.9949.

*3-chloro-5-(trifluoromethyl)-2-(4-((4-((trifluoromethyl)sulfinyl)benzyl)oxy)phenyl)pyridin (**6a**)*: White solid, yield 41.7%, m.p. 123.3–124.7 °C. ^1^H NMR (400 MHz, DMSO-d_6_) δ: 9.00 (s, 1H), 8.53 (s, 1H), 7.94 (d, J = 8.1 Hz, 2H), 7.82 (d, J = 8.3 Hz, 2H), 7.76 (d, J = 8.7 Hz, 2H), 7.19 (d, J = 8.8 Hz, 2H), 5.35 (s, 2H). ^13^C NMR (101 MHz, DMSO-d_6_) δ: 159.17, 158.87, 144.40 (q, J = 3.7 Hz), 143.20, 135.85 (q, J = 3.4 Hz), 134.70 (q, J = 1.4 Hz), 131.14, 129.50, 129.28, 128.67, 126.26, 124.83 (q, J = 337.5 Hz), 124.47 (q, J = 32.9 Hz), 122.96 (q, J = 273.7 Hz), 114.45, 68.47. HRMS (ESI): calculated for C_20_H_13_ClF_6_NO_2_S [M+H]^+^ 480.0254, 482.0225, found 480.0254, 482.0227.

*3-chloro-2-(3-fluoro-4-((4-((trifluoromethyl)sulfinyl)benzyl)oxy)phenyl)-5-(trifluoromethyl)pyridine (**6b**)*: White solid, yield 80.3%, m.p. 113.0–115.0 °C. ^1^H NMR (400 MHz, DMSO-d_6_) δ: 9.02 (s, 1H), 8.57 (s, 1H), 7.97 (d, J = 7.8 Hz, 2H), 7.84 (d, J = 8.0 Hz, 2H), 7.69 (dd, J = 12.2, 1.5 Hz, 1H), 7.62 (d, J = 8.2 Hz, 1H), 7.43 (t, J = 8.6 Hz, 1H), 5.45 (s, 2H). ^13^C NMR (101 MHz, DMSO-d_6_) δ: 157.59, 150.96 (d, J = 245.2 Hz), 147.10 (d, J = 10.4 Hz), 144.36 (q, J = 3.7 Hz), 142.56, 135.91 (q, J = 3.2 Hz), 134.94 (q, J = 1.0 Hz), 129.88 (d, J = 6.6 Hz), 129.44, 128.68, 126.31, 126.25, 124.86 (q, J = 33.1 Hz), 124.77 (q, J = 337.7 Hz), 122.81 (q, J = 273.9 Hz), 117.26 (d, J = 19.9 Hz), 114.75, 69.38. HRMS (ESI): calculated for C_20_H_10_ClF_7_NO_2_S [M−H]^−^ 496.0014, 497.9985, found 496.0015, 497.9993.

*3-chloro-2-(3-chloro-4-((4-((trifluoromethyl)sulfinyl)benzyl)oxy)phenyl)-5-(trifluoromethyl)pyridine (**6c**)*: White solid, yield 50.1%, m.p. 119.3–120.3 °C. ^1^H NMR (400 MHz, DMSO-d_6_) δ: 9.01 (s, 1H), 8.58 (s, 1H), 7.96 (d, J = 8.1 Hz, 2H), 7.87–7.83 (m, 3H), 7.76 (dd, J = 8.6, 1.9 Hz, 1H), 7.40 (d, J = 8.7 Hz, 1H), 5.46 (s, 2H). ^13^C NMR (101 MHz, DMSO-d_6_) δ: 157.55, 154.35, 144.47 (q, J = 3.9 Hz), 142.68, 135.98 (q, J = 3.3 Hz), 134.86 (q, J = 1.0 Hz), 131.02, 130.37, 129.78, 129.53, 128.44, 126.33, 124.93 (q, J = 33.2 Hz), 124.83 (q, J = 337.6 Hz), 122.88 (q, J = 274.1 Hz), 121.23, 113.67, 69.30. HRMS (ESI): calculated for C_20_H_10_Cl_2_F_6_NO_2_S [M−H]^−^ 511.9719, 513.9689, found 511.9718, 513.9692.

*3-chloro-2-(3-nitro-4-((4-((trifluoromethyl)sulfinyl)benzyl)oxy)phenyl)-5-(trifluoromethyl)pyridine (**6d**)*: Yellow solid, yield 96.2%, m.p. 116.0–118.0 °C. ^1^H NMR (400 MHz, DMSO-d_6_) δ: 9.09 (s, 1H), 8.66 (s, 1H), 8.41 (d, J = 2.1 Hz, 1H), 8.17 (dd, J = 8.8, 2.1 Hz, 1H), 8.01 (d, J = 8.1 Hz, 2H), 7.87 (d, J = 8.2 Hz, 2H), 7.67 (d, J = 8.9 Hz, 1H), 5.61 (s, 2H). ^13^C NMR (101 MHz, DMSO-d_6_) δ: 156.67, 151.67, 144.54 (q, J = 3.6 Hz), 142.05, 138.94, 136.03 (q, J = 3.5 Hz), 135.44, 134.97, 129.67, 129.31, 128.30, 126.38, 126.28, 125.30 (q, J = 33.3 Hz), 124.77 (q, J = 337.8 Hz), 122.76 (q, J = 274.1 Hz), 115.28, 69.86. HRMS (ESI): calculated for C_20_H_10_ClF_6_N_2_O_4_S [M−H]^−^ 522.9959, 524.9930, found 522.9961, 524.9921.

*3-chloro-5-(trifluoromethyl)-2-(3-(trifluoromethyl)-4-((4-((trifluoromethyl)sulfinyl)benzyl)oxy)phenyl)pyridine (**6e**)*: White solid, yield 51.1%, m.p. 117.9–120.1 °C. ^1^H NMR (400 MHz, DMSO-d_6_) δ: 9.05 (s, 1H), 8.60 (s, 1H), 8.09 (d, J = 9.8 Hz, 2H), 7.98 (s, 2H), 7.82 (d, J = 5.1 Hz, 2H), 7.54 (d, J = 6.4 Hz, 1H), 5.55 (s, 2H). ^13^C NMR (101 MHz, DMSO-d_6_) δ: 157.40, 156.58, 144.48 (q, J = 2.7 Hz), 142.44, 135.95, 135.37, 134.86 (q, J = 1.5 Hz), 129.53, 129.15, 128.20, 128.09, 126.29, 125.04 (q, J = 33.2 Hz), 124.77 (q, J = 339.3 Hz), 123.44 (q, J = 273.4 Hz), 122.80 (q, J = 274.3 Hz), 117.06 (q, J = 30.8 Hz), 113.78, 69.23. HRMS (ESI): calculated for C_21_H_10_ClF_9_NO_2_S [M−H]^−^ 545.9983, 547.9953, found 545.9985, 547.9960.

*3-chloro-2-(2,3-difluoro-4-((4-((trifluoromethyl)sulfinyl)benzyl)oxy)phenyl)-5-(trifluoromethyl)pyridine (**6f**)*: White solid, yield 48.4%, m.p. 122.4–24.4 °C. ^1^H NMR (500 MHz, CDCl_3_) δ: 8.79-8.75 (m, 1H), 7.99 (d, J = 1.5 Hz, 1H), 7.78 (d, J = 8.2 Hz, 2H), 7.66 (d, J = 8.5 Hz, 2H), 7.13–7.10 (m, 1H), 6.86–6.83 (m, 1H), 5.25 (s, 2H). ^13^C NMR (126 MHz, CDCl_3_) δ: 154.15, 148.13 (dd, J = 253.0, 11.6 Hz), 147.93 (dd, J = 8.1, 3.1 Hz), 143.30 (q, J = 3.9 Hz), 140.90, 140.59 (dd, J = 250.4, 14.4 Hz), 134.73 (q, J = 1.5 Hz), 133.79 (q, J = 3.5 Hz), 131.19, 127.08, 126.08 (q, J = 33.9 Hz), 125.42, 123.76 (t, J = 4.0 Hz), 123.66 (q, J = 335.8 Hz), 121.56 (q, J = 273.6 Hz), 119.28 (d, J = 12.4 Hz), 109.16 (d, J = 3.1 Hz), 69.60. HRMS (ESI): calculated for C_20_H_9_ClF_8_NO_2_S [M−H]^−^ 513.9920, 515.9891, found 513.9916, 515.9887.

*3-chloro-5-(trifluoromethyl)-2-(4-((4-((trifluoromethyl)sulfonyl)benzyl)oxy)phenyl)pyridine (**7a**)*: White solid, yield 40.1%, m.p. 132.7–134.8 °C. ^1^H NMR (500 MHz, DMSO-d_6_) δ: 9.02–8.97 (m, 1H), 8.53 (d, J = 1.5 Hz, 1H), 8.19 (d, J = 8.4 Hz, 2H), 7.94 (d, J = 8.5 Hz, 2H), 7.78–7.75 (m, 2H), 7.21–7.18 (m, 2H), 5.45 (s, 2H). ^13^C NMR (126 MHz, DMSO-d_6_) δ: 158.92, 158.78, 147.74, 144.32 (q, J = 3.8 Hz), 135.76 (q, J = 3.5 Hz), 131.12, 131.03, 129.65, 129.23, 128.92, 128.58, 124.46 (q, J = 32.9 Hz), 122.88 (q, J = 273.3 Hz), 119.41 (q, J = 326.7 Hz), 114.42, 68.10. HRMS (ESI): calculated for C_20_H_11_ClF_6_NO_3_S [M−H]^−^ 494.0058, 496.0028, found 494.0061, 496.0035.

*3-chloro-2-(3-fluoro-4-((4-((trifluoromethyl)sulfonyl)benzyl)oxy)phenyl)-5-(trifluoromethyl)pyridine (**7b**)*: White solid, yield 48.4%, m.p. 111.8–114.0 °C. ^1^H NMR (400 MHz, DMSO-d_6_) δ: 9.02 (s, 1H), 8.58 (s, 1H), 8.23 (d, J = 8.2 Hz, 2H), 7.95 (d, J = 8.4 Hz, 2H), 7.71 (dd, J = 12.3, 1.9 Hz, 1H), 7.62 (d, J = 8.5 Hz, 1H), 7.41 (t, J = 8.7 Hz, 1H), 5.54 (s, 2H). ^13^C NMR (101 MHz, DMSO-d_6_) δ: 157.57, 150.93 (d, J = 245.4 Hz), 147.12, 146.91 (d, J = 10.5 Hz), 144.37 (q, J = 3.9 Hz), 135.92 (q, J = 3.4 Hz), 131.11, 130.09 (d, J = 6.8 Hz), 129.47, 128.96, 128.79, 126.34 (d, J = 3.2 Hz), 124.90 (q, J = 33.3 Hz), 122.81 (q, J = 274.0 Hz), 119.41 (q, J = 327.6 Hz), 117.33 (d, J = 19.8 Hz), 114.76, 68.99. HRMS (ESI): calculated for C_20_H_10_ClF_7_NO_3_S [M−H]^−^ 511.9964, 513.9934, found 511.9968, 513.9937.

*3-chloro-2-(3-chloro-4-((4-((trifluoromethyl)sulfonyl)benzyl)oxy)phenyl)-5-(trifluoromethyl)pyridine (**7c**)*: White solid, yield 68.9%, m.p. 142.6–144.8 °C. ^1^H NMR (500 MHz, DMSO-d_6_) δ: 9.04–9.00 (m, 1H), 8.57 (d, J = 1.5 Hz, 1H), 8.24 (d, J = 8.4 Hz, 2H), 7.97 (d, J = 8.5 Hz, 2H), 7.90 (d, J = 2.2 Hz, 1H), 7.78 (dd, J = 8.6, 2.2 Hz, 1H), 7.40 (d, J = 8.7 Hz, 1H), 5.57 (s, 2H). ^13^C NMR (126 MHz, DMSO-d_6_) δ: 157.43, 154.12, 147.14, 144.36 (d, J = 3.7 Hz), 135.86 (d, J = 3.3 Hz), 131.07, 130.97, 130.50, 129.71, 129.45, 128.74, 128.66, 124.91 (q, J = 33.3 Hz), 122.78 (q, J = 273.4 Hz), 121.23, 119.39 (q, J = 326.9 Hz), 113.64, 68.94. HRMS (ESI): calculated for C_20_H_10_Cl_2_F_6_NO_3_S [M−H]^−^ 527.9668, 529.9639, found 527.9664, 529.9640.

*3-chloro-2-(3-nitro-4-((4-((trifluoromethyl)sulfonyl)benzyl)oxy)phenyl)-5-(trifluoromethyl)pyridine (**7d**)*: Yellow solid, yield 24.8%, m.p. 129.1–131.6 °C. ^1^H NMR (500 MHz, DMSO-d_6_) δ: 9.06 (d, J = 0.9 Hz, 1H), 8.62 (d, J = 1.4 Hz, 1H), 8.40 (d, J = 2.3 Hz, 1H), 8.25 (d, J = 8.4 Hz, 2H), 8.15 (dd, J = 8.8, 2.3 Hz, 1H), 7.95 (d, J = 8.4 Hz, 2H), 7.63 (d, J = 8.9 Hz, 1H), 5.67 (s, 2H). ^13^C NMR (126 MHz, DMSO-d_6_) δ: 156.61, 151.50, 146.52, 144.51 (q, J = 3.8 Hz), 138.85, 136.00 (q, J = 3.6 Hz), 135.50, 131.09, 129.65, 129.47, 128.86, 128.61, 126.43, 125.30 (q, J = 33.3 Hz), 122.72 (q, J = 273.5 Hz), 119.38 (q, J = 326.8 Hz), 115.26, 69.51. HRMS (ESI): calculated for C_20_H_10_ClF_6_N_2_O_5_S [M−H]^−^ 538.9909, 540.9879, found 538.9905, 540.9882.

*3-chloro-5-(trifluoromethyl)-2-(3-(trifluoromethyl)-4-((4-((trifluoromethyl)sulfonyl)benzyl)oxy)phenyl)pyridine (**7e**)*: White solid, yield 35.2%, m.p. 111.2–113.4 °C. ^1^H NMR (400 MHz, DMSO-d_6_) δ: 9.05 (s, 1H), 8.61 (s, 1H), 8.26 (d, J = 8.1 Hz, 2H), 8.13 (d, J = 9.0 Hz, 1H), 8.09 (s, 1H), 7.93 (d, J = 8.2 Hz, 2H), 7.53 (d, J = 8.7 Hz, 1H), 5.65 (s, 2H). ^13^C NMR (101 MHz, DMSO-d_6_) δ: 157.39, 156.37, 146.94, 144.49 (q, J = 3.6 Hz), 135.95 (q, J = 3.4 Hz), 135.44, 131.17, 129.55, 129.34, 128.77, 128.46, 128.22 (q, J = 5.3 Hz), 125.07 (q, J = 33.0 Hz), 123.44 (q, J = 273.5 Hz), 122.80 (q, J = 274.1 Hz), 119.41 (q, J = 327.3 Hz), 117.04 (q, J = 30.7 Hz), 113.77, 68.90. HRMS (ESI): calculated for C_21_H_10_ClF_9_NO_3_S [M−H]^−^ 561.9932, 563.9902, found 561.9935, 563.9913.

*3-chloro-2-(2,3-difluoro-4-((4-((trifluoromethyl)sulfonyl)benzyl)oxy)phenyl)-5-(trifluoromethyl)pyridine (**7f**)*: White solid, yield 37.7%, m.p. 102.8–104.8 °C. ^1^H NMR (500 MHz, CDCl_3_) δ: 8.77 (d, J = 1.1 Hz, 1H), 8.02 (d, J = 8.3 Hz, 2H), 7.99 (d, J = 1.6 Hz, 1H), 7.72 (d, J = 8.4 Hz, 2H), 7.15–7.11 (m, 1H), 6.87–6.82 (m, 1H), 5.29 (s, 2H). ^13^C NMR (126 MHz, CDCl_3_) δ: 155.22, 149.30 (dd, J = 253.0, 11.3 Hz), 148.83 (dd, J = 8.2, 3.1 Hz), 145.53, 144.47 (q, J = 3.8 Hz), 141.75 (dd, J = 250.2, 14.3 Hz), 134.96 (q, J = 3.5 Hz), 132.35, 131.42, 131.27, 128.14, 127.29 (q, J = 33.9 Hz), 124.99 (t, J = 3.9 Hz), 122.70 (q, J = 273.6 Hz), 119.92 (q, J = 326.4 Hz), 120.71 (d, J = 12.5 Hz), 110.29 (d, J = 3.1 Hz), 70.38. HRMS (ESI): calculated for C_20_H_9_ClF_8_NO_3_S [M−H]^−^ 529.9869, 531.9840, found 529.9869, 531.9824.

### 3.3. Herbicidal Activity Test

Levels of herbicidal activity for compounds **5a**–**5f**, **6a**–**6f**, and **7a**–**7f** against the monocotyledonous weeds Digitaria sanguinalis (DS), Echinochloa crusgalli (EC), and Setaria viridis (SV), and the dicotyledonous weeds Abutilon theophrasti (AT), Amaranthus retroflexus (AR), and Eclipta prostrate (EP) were determined using previously disclosed methods [29,30,31,34].

All target compounds were dissolved in *N*,*N*-dimethylformamide, diluted to an appropriate dose with distilled water containing 0.1% Tween-80 emulsifier, and used for seedlings at the three-leaf stage with three replicates. Seedlings treated with *N*,*N*-dimethylformamide and distilled water containing 0.1% Tween-80 emulsifier served as blank controls, with fomesafen-treated seedlings serving as positive controls. The application rates were 150, 75, and 37.5 g a.i./ha. Herbicidal activity was assessed visually after a 20-day period, with the results being listed in Table 1.

## 4. Conclusions

In conclusion, 18 novel α-trifluorothioanisole derivatives containing phenylpyridine moieties were prepared as candidate herbicides. The preliminary herbicidal activity assay results show that some target compounds exhibited good herbicidal activities against broadleaf weeds at 150 g a.i./hm^2^. Among these, compound **5a** possessed excellent activity (>85%) against AT, AR, and EP at 37.5 g a.i./hm^2^, which was slightly superior to fomesafen. Furthermore, compound **5f** possessed good activity (>70%) against AT, AR, and EP at 37.5 g a.i./hm^2^, which was equivalent to that of fomesafen. Thus, compound **5a** may be a lead compound for further structural optimization.

## Data Availability

Samples of the compounds are not available from the authors.

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
