# Peer review of "Synthesis of Novel α-Trifluorothioanisole Derivatives Containing Phenylpyridine Moieties with Herbicidal Activity"

_molecules, 2022, doi:10.3390/molecules27185879_

Round 1
Reviewer 1 Report
It is very interesting work. Written carefully. The compounds are new and well described.
This work can be published after minor corrections.
Minors:
Page 3, line 60-61; “while compounds 6e and 7f exhibited > 80% activity against AT and AR.” It obviously the above description of 6e is out of accord with the date in Table 1. 6e may be 6f?
Page 3, line 66; where is 5h come from?
Page 5-8, Scheme 1; almost all target compounds contain Cl, there should be listed isotope peak date in the HRMS date.
Reviewer 2 Report
This manuscript by Cai et al. describes “Synthesis of Novel α-Trifluorothioanisole Derivatives Containing Phenylpyridine Moieties with Herbicidal Activity”. This is a nice extension of their previous paper published in the J. Heterocycl. Chem. 2022, 59, 1247-1252. This article is well written, all compounds are novel, and well characterized with 1H NMR, 13C NMR, and HRMS. One of the compounds was also characterized through X-ray crystallography to confirm structure. Some compounds displayed superior activity in compared to positive control fomesafen at a dose of 37.5 g a.i./hm2. I would recommend this manuscript to be published after minor comments as below
1. In introduction section, please cite and write about some relevant published papers on small molecules having herbicidal activity ( J. Pestic. Sci. 41(4), 133–144 (2016); J. Agric. Food Chem. 2021, 69, 30, 8415–8427; J. Agric. Food Chem. 2020, 68, 15107−15114.)
2. In section 2.2, authors should discuss structure-activity relationships of various groups on herbicidal activity.
3. Compound 7a has 12 protons in its structure but it written 10 in its characterization data and correct it in manuscript (page 8) as well integrate corresponding peak in supplementary information. Calculated formula [M-H]- for 6c should be C20H10Cl2F6NO2S (page 7; line 188), compound 6d calculated formula [M-H]- should have H9 instead of H10 (page 7, line 197), calculated [M-H]- formula for 7c should be C20H10Cl2F6NO3S (page 8, line 238); same errors should be checked again through whole manuscript and should be corrected in final version.
4. Authors should write procedure performed for herbicidal assay after citing reference in section 3.3.
Reviewer 3 Report
This paper is clearly written and well organized and. The introduction and background are reasonable given the premise of the paper. I appreciate authors for screening of herbicidal activities with different types of derivative compounds. I have comment included for the betterment of the paper
Can authors explain the why trifluoromethylthio group showing more potent activity then the corresponding its oxidized functional groups.
